# Urban Sustainability at the Cost of Rural Unsustainability

**Guangwei Huang**

Graduate School of Global Environmental Studies, Sophia University, Tokyo 102-8554, Japan;
huanggwx@sophia.ac.jp; Tel.: +81-3-3238-4667

**Abstract:** Urban sustainability refers to building and maintaining cities that can continue to function without running out of resources. However, growing cities require more land and urban sprawl has transformed surrounding rural areas into urbanized settlements. Furthermore, the prosperity of large cities depends on the supply of both natural and human resources from rural areas, either nearby or remote. On the other hand, the use of resources of rural areas by cities may cause negative externalities to rural areas, affecting their sustainability. Therefore, a critical, but very much neglected issue, is how unban sustainability should be pursued without affecting rural sustainability. In this study, cases in Japan and China were analyzed from resources and population migration perspectives to provide evidence for the possibility that urban sustainability might have been pursued at the cost of rural unsustainability. It was intended to develop a better understanding of urban sustainability through the lens of externalities. Based on the analysis, a new framework for urban sustainability study was proposed, which consists of three new pillars. Namely, externality, vulnerability, and population instability.

**Keywords:** urban; rural; externality; population; water; food; energy; framework

## 1. Introduction

Sustainability is both a simple and complex concept. It is simple because the word is straight and self-explaining; it is complex because the true meaning is about the integration of environmental conservation, social equity, and economic vitality to create thriving, healthy, diverse, and resilient communities for this generation and generations to come. Therefore, the practice or pursuit of sustainability should recognize how these issues are interconnected and require a systems approach to deal with the complexity. The most widely used definition comes from the UN World Commission on Environment and Development: "sustainable development is the development that meets the needs of the present without compromising the ability of future generations to meet their own needs." Although well-articulated, this definition may cause misunderstanding that sustainability is mainly concerned with the intergenerational relationship because its rhetorical accent appears to be placed on intergenerational justice. As a matter of fact, the fair use of natural and human resources and fair distribution of wealth among the present generation of people or efforts towards fair use and distribution should be understood as a presumption to the most quoted definition of sustainability. However, the reality is that there are various disparities in today's world, such as disparities in resource use, benefits sharing, health care, and basic education. Even in the United States, 11.8% of the population or 38.1 million people still live in relative poverty ($33.26 per day), and the poverty rate for female-householder families reached 24.9% in 2018 [1]. Therefore, the prosperity for the present generation without anyone being left behind should be more emphasized in sustainability study, and research efforts to find out bottlenecks in achieving the objective should be further promoted. In consideration of the current development context, a focus that sustainability science has been placed on is the rapid and global increase of urbanization, which may be viewed either as an opportunity or a challenge to society. Over the past several decades, many urban sustainability studies were conducted targeting urbanized

areas ranging from large ones, such as London [2], Tokyo [3], Hong Kong [4], Beijing, and Shanghai [5] to medium and small cities, such as Rotterdam [6], Simao county [7], and even down to district or community level [8]. However, as pointed out by Huang [9], those studies looked at sustainable development in an isolated manner or at a single scale so that they failed to address the possibility that the sustainability of a city may become the cause of un-sustainability of other cities, or some cities may be sacrificed for a particular city being "sustainable".

In the 1960s, the core-periphery theory was developed by sociologists and economists in view of the relative development lag in the Latin America continent [10]. According to the theory, the core is part of a country where economic activities and development are the most, while the periphery is the area of low or declining economic development within a country. This concept has gained a large following in social, political science, and economics fields since the 1970s. Consequently, a large number of studies have led to the development of New Economic Geography as a major field of economics to explain the spatial economic structure [11,12]. Yet most studies in New Economic Geography have remained confined to two-city models, in which spatial economic concentration to a single city is triggered by bifurcation [13,14]. Thus, the theory is less relevant to the current development in advanced economies. More importantly, the core-periphery theory has not been examined from a sustainability perspective. What is missing in the theory is the accounting for environmental loading a core may cause to its periphery.

A new concept to urban planning and development is the Doughnut economics framework, which combines the concept of planetary boundaries with the complementary concept of social boundaries [15]. It provides a new platform to think about how to solve environmental and socio-economic challenges in a coherent and balanced way and it has been incorporated in Amsterdam's circular economy strategy. However, to downscale the concept to a city level, methodology development is needed.

A related concept is the urban metabolism, proposed by Wolman in 1965, who viewed the urban system as closely resembling an organism with the equivalent of metabolic processes [16]. It is focused on inputs of materials and energy into the urban system and outputs of wastes from the system. Assessing the potential impacts of economic activities on urban metabolic components can provide a quantitative estimation of how a city depends on the external. However, urban metabolic studies have so far failed to address the question of how increasing the metabolism of a city might affect the sustainability of other cities or regions that supply materials and energy to the city. Besides, the degree of urban sustainability cannot be simply judged by materials and energy flow [17].

Driven by the Sustainable Development Goals of the UN, urban sustainability studies were further accelerated since 2015. Especially, Goal 11 is to make cities and human settlements inclusive, safe, resilient, and sustainable. Under the goal, Target 11.a is "Support positive economic, social and environmental links between urban, peri-urban and rural areas by strengthening national and regional development planning", which promotes studies on relationships between urban and immediately adjoining areas. However, this target also fails to explicitly address the need to consider the possible connection between the prosperity of a city and the decay of another city far away. An obvious example is the city of Detroit, which had gone through a major economic and demographic decline in the 1980s, due to international competition, particularly from Japanese and German automobile makers. With the business success of Toyota Motor Corporation, Toyota City (where the headquarter of Toyota Motor Corporation is located) was ranked the happiest place for living among 48 medium-sized Japanese cities in 2018 based on various indicators from job opportunity to health [18]. A research initiative deserving mentioning is the development of The Arcadis Sustainable Cities Index [19], which employs information on three pillars of sustainability: People (social), Planet (environmental), and Profit (economic) to rank global cities. In its methodology, although there is a demographics indicator, population migration is not considered explicitly. Similarly, the energy indicator includes energy use but does not consider where energy is generated. In view of these missing or insufficiently

analyzed links between a city and the rest of the world in existing studies, the present study was motivated to shed new light on environmental conflicts between a city and the regions that supply materials and energy to the city and propose a new concept for better understanding of urban sustainability.

## 2. Materials and Methods

To obtain a broad appreciation of the need and importance of considering both intergenerational and interregional fairness, the case study approach was employed because it can provide concrete, in-depth, and multi-angle views of a complex issue in its real-life context. The case study approach may also be referred to as a "naturalistic" research design in contrast to an "experimental" design because a case study is both the process of learning about the case and the product of our learning as stated by Stake [20]. Furthermore, the case study approach can be designed to answer "how", "what", and "why" questions, lending itself well to exploring the key characteristic of a case, elucidating causal links and pathways resulting from a new policy initiative or a development project. Therefore, the use of the case study approach for the present study is justifiable.

The steps of a case study are (1) defining the case; (2) selecting the case(s); (3) collecting and analyzing the data; (4) interpreting data; and (5) reporting the findings. Unlike a social opinion survey, random selection is not required. Instead, unusual, neglected, or outlying cases are sought to highlight the research problem.

In this study, we defined cases as those having urban environmental and social externalities. In economics, the term externality refers to activities of individuals (or firms) that affect other non-involved individuals [21]. In this study, we define urban environmental and social externalities as an environmental and social cost or benefit of urban development activity experienced by a third party. For example, a housing development in a large city used wood from a small city with forests. Due to unequal status and the lack of knowledge on marketing and ecological importance of forests, the use of forests and logging labor work of the small city was insufficiently compensated, which resulted in negative environmental and social externalities.

To select cases, we targeted cities in China and Japan. The reason is that Japan had rapid urbanization during the 1960s and 1970s, and China is experiencing superfast urbanization in recent decades. In 1980, less than 20% of the Chinese population lived in cities, but in 2019, the urban population in China reached 60% [22]. The selected cases are paired having one megacity with negative externalities to other regions exporting ecosystem services to that megacity, while having negative externalities to itself. Data for study cases were collected from statistical annual reports by various government entities and from literature as well. Although data are secondary, the data blending was designed for elucidating environmental conflicts between the pair of targets. For instance, by identifying Tokyo-induced water use in other regions and examining environmental issues faced by these regions, causes of environmental issues can be explained and more effective countermeasures can be proposed. It should be mentioned here that the case studies in the present work are not aimed at being comprehensive but illustrative and persuasive.

The footprint of a city is defined as the amount of land required to sustain its metabolism; that is, to provide the raw materials on which it feeds, and process the waste products it excretes [23]. It has been used to assess the impact of cities on the environment. For example, the ecological footprint of London, UK is estimated to be 120 times the area of the city itself [24]. Such an ecological deficit of a city means that the city is either importing biocapacity through trade or liquidating regional ecological assets. Although footprint can be used to estimate the maximum "earth share" a city can use without depriving either future generations or those now living in other regions of the world, it does not provide any concrete information about which city is sacrificed for which city.

On the other hand, sustainability is not just determined by resource utilization. Cities without having ecological deficit are not guaranteed to be sustainable. Population dynamism is also a determinant factor for urban sustainability but has not been explicitly

taken into consideration in the ecological footprint framework. Besides, the ecological footprint is an aggregate indicator, adding up all the productive areas on which a population or a person depends on. For a better understanding of resource use and resultant externality, the breaking down of ecological footprint into its constituent parts (landfilled waste, energy, transport, food, water, etc.) is desirable and the dependence of water, energy, and food of a city on other cities or regions should be investigated and used for assessing urban sustainability.

Over the past decades, cities such as New York, Chicago, Birmingham, Glasgow, Budapest, and Yubari have been losing residents [25]. Cities with a depopulation trend may suffer from falling property values, declining tax revenues, and increasing unemployment while having to maintain water and electricity supply, roads, schools, and other public services that are overcapacity for the current population size. As a result, depopulation will cause economic stress and affect resident's well-being. On the other hand, overpopulation in a large city will cause many problems ranging from traffic congestion, urban poverty to disaster risk increase. Therefore, population stability can be considered as a surrogate for well-being and then used as an indicator for urban sustainability at least for developed and emerging countries.

Following this line of thinking, this study assessed urban sustainability in terms of self-sufficiency rates of water, energy, and food. Above these, it examined the population movement as pivotal to characterize the externality of urban development. Based on case studies, a new framework is proposed for the evaluation of urban sustainability.

## 3. Results

### 3.1. Case 1: Tokyo-Niigata

Figure 1 shows the map of Japan indicating the locations of Tokyo and Niigata. Tokyo is one of the world's largest urban economies by gross domestic product and is the main engine of the Japanese economy.

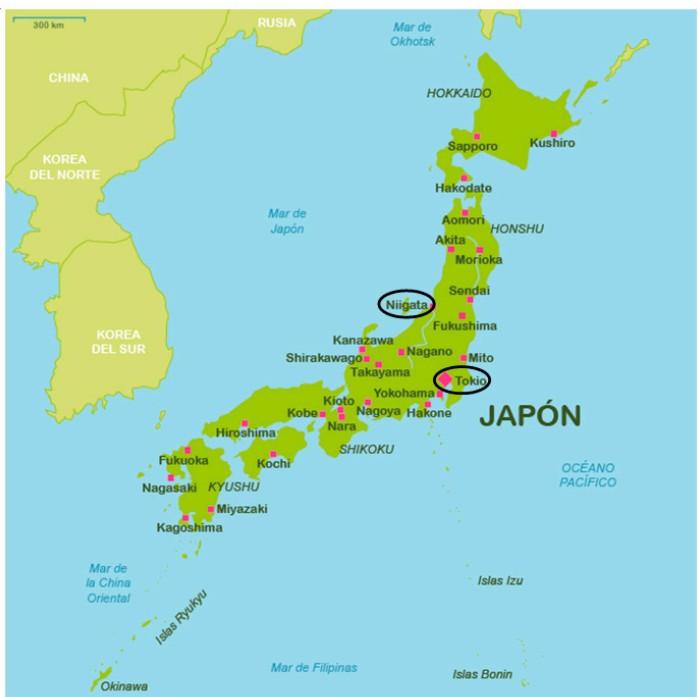

**Figure 1.** Locations of Tokyo and Niigata Prefecture.

A critical issue in Japan is that its population has been shrinking since 2007. In Japan, the total fertility rate—the average number of babies a woman gives birth to during her life—has fallen from 2.08 in 1974 to 1.36 in 2019 [26], while a total fertility rate of 2.07 is

required to maintain its population levels [27]. Because Japan's economy and social systems were structured on the premise of population growth, the depopulation will profoundly impact the workforce, making it difficult to maintain the current social security systems, such as pension benefits and medical insurance. Tokyo, however, is an exception among Japanese cities to the depopulation trend. As shown in Figure 2, Tokyo's population was very stable during the 1980s and the 1990s and showed an increasing trend since 2000. Nevertheless, the number of births per year has dropped so largely since 2000 as compared to previous decades. Tokyo's fertility rate declined from 2.73 in 1950 to 1.23 in 1990 and to its lowest 1.0 in 2005, which is worse than the national average [28]. Therefore, the population growth in Tokyo is maintained by the population influx from other regions.

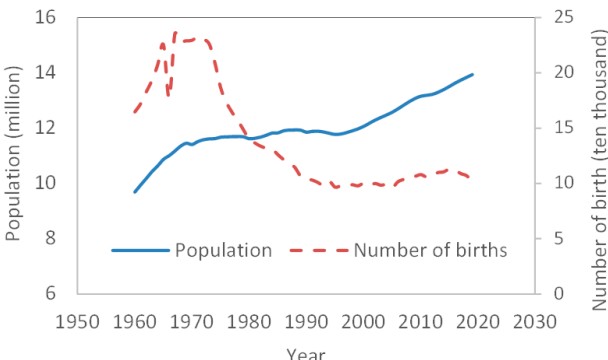

**Figure 2.** The population and number of births in Tokyo (Data source: [28]).

The Japanese government has been promoting the revitalization of local economies with many initiatives in recent years. The annual regional revitalization budget in recent years is 100 billion JPY. There is an annual grant for individual measures based on the comprehensive strategy in addition to the regional revitalization budget. The amount is more than 650 billion JPY in 2017 and increased to one trillion yen in 2020.

However, government-sponsored programs have not produced tangible results. In particular, they failed to achieve the goal of striking a balance between population inflows into the Tokyo metropolitan area and outflows from other regions across the country. People who moved into Tokyo and three surrounding prefectures outnumbered those who moved out of the metropolitan area by some 86,000 in 2019, indicating that Tokyo's population is still increasing. Besides, more females migrated to Tokyo than males, and most of the migrants are in the age group of 20–40 [29]. Since the conditions for childrearing in Tokyo is not favorable, as evidenced by the large number of children waiting to enter kindergarten, many women in Tokyo might not want to have children as often discussed in social media. Therefore, the concentration of females in the capital might cause the national birth rate to decline further.

One of the population influx sources is the Tohoku region in Japan. The total population is predicted by the National Institute of Population and Social Security Research of Japan to drop from 9.33 million today to 6.86 million by 2040, and then to just over 5 million by 2060 [30]. Aomori and Akita will be hit particularly hard. The population of Aomori is expected to drop from around 1.3 million now to 932,000 by 2040, and Akita is expected to drop from around 1 million to 700,000. The number of women between 20 and 40 in both prefectures will drop by 48 percent by 2040. These data indicate again that the current prosperity of Tokyo has a cost of the unsustainability of other cities.

Using interregional input-output table and water consumption data, a study by Ishiro [31] revealed Tokyo's dependence of water on other regions, as shown in Figure 3. As seen clearly from the figure, for the water use in Tokyo, local water resources provided 700 million m$^3$ while Ibaraki and Chiba prefectures supplied 800 and 500 million m$^3$, respectively. In total, other prefectures covered 80% of the water demand in Tokyo. Wording differently, the water self-sufficiency rate in Tokyo is 20%. Among regions contributing

water resources to Tokyo, Niigata Prefecture is about 300 km away from Tokyo and is the farthest among them.

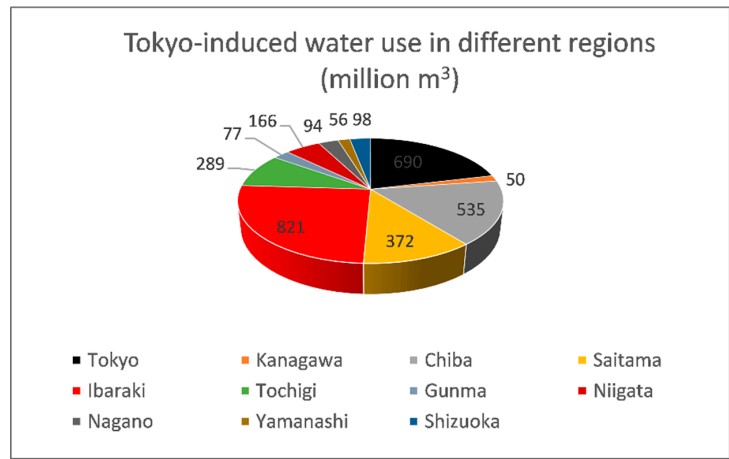

**Figure 3.** Tokyo's dependence on other regions in Japan for water resources (Data source: [31]).

According to the estimation by the Ministry of Agriculture and Forest, Japan, the food self-sufficiency rate of Tokyo is as low as 1% on Calorie base and 3% on production base [32]. Using the inter-prefecture material flow tables compiled by the Ministry of Land, Infrastructure, Transport and Tourism, Japan, Niigata Prefecture contributes about 3% of food consumed in Tokyo in terms of weights of nine food items from rice, beans, millet to seafood exported from Niigata to Tokyo. On the other hand, according to the Bureau of Environment, Tokyo Metropolitan Government, the electricity self-sufficiency rate in Tokyo is 11% [33]. Most of the electricity used in Tokyo was generated in other prefectures, including Niigata.

Niigata Prefecture stretches about 240 km along the Sea of Japan and consists of 12 cities. It is the fifth-largest prefecture of Japan in terms of land surface area. As revealed by the land use map (Figure 4), the primary industry in Niigata Prefecture is agriculture, with rice as its principal product. It has the highest rice output among rice-producing prefectures of Japan, covering 8.1% of Japan's rice market. Niigata's food self-sufficiency rate is 107% on calorie base and 108% based on production, respectively. In addition to water resources and food, Tokyo also uses energy produced in Niigata. The Kashiwazaki-Kariwa Nuclear Power Plant is in Niigata Prefecture, which started in 1993. It is the largest nuclear generating station in the world by net electrical power rating, owned and operated by Tokyo Electric Power Company (TEPCO) and supplies electricity to Tokyo.

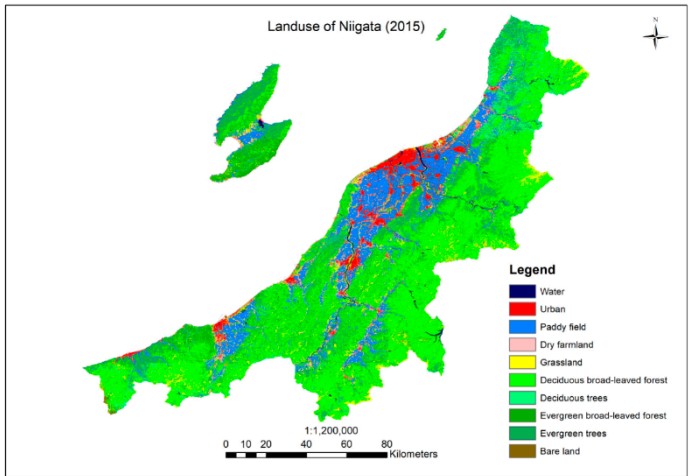

**Figure 4.** The land use in Niigata Prefecture (Data source: [34]).

Tokyo is well known as having one of the most efficient and convenient public transport systems in the world. It combines an extensive train network, subway lines, and a bus system. The percentage of public transport use in Tokyo is 65%. One of Tokyo's busiest and most important lines is the Yamanote Line, a railway loop line connecting most of Tokyo's major stations and urban centers, such as Marunouchi, Yurakucho, Shinagawa, Shibuya, and Shinjuku. During peak times, the electricity for running the train is generated from a hydro-power station in the Shinano River's middle reaches, which is in the Niigata Prefecture. A study by Huang [35] reported that the hydro-power station's operation caused thermal anomaly in the river reaches downstream of the hydro-power station. At an immediate downstream site, the diurnal variation of water temperature exhibited two peaks. The first peak was ahead of the air temperature peak in the morning due to water release from the power plant. At another site 17 km downstream of the power plant, daily maximum water temperature often occurred around midnight, due to heat transport from the upstream. Besides, the hydro-power generation operation also altered the flow regime, forming a discharge reduction reach of 63.5 km. The use of public transportation reduces $CO_2$ emission but is not free of other environmental loadings. The sustainable transport system in Tokyo carries an environmental externality to the waterway in Niigata Prefecture.

As shown in Figure 5, the population of Niigata Prefecture has been declining steadily. In 2019, among those moving out of Niigata, 49.1% settled down in Tokyo, according to the Statistics Bureau of Japan's data [36]. As the population declined, the sector being hit first was agriculture. According to the data compiled by the Ministry of Agriculture, Forest and Fishery, Japan, the number of persons engaged in agriculture in Niigata Prefecture decreased from 246,019 in 2005 to 148,941 in 2015. Also shown in Figure 5, the GDP generated from Niigata's agricultural sector has been decreasing in recent years, due partially to the population decrease [37]. With the continuous movement of both natural and human resources from Niigata to Tokyo, Niigata's sustainability will be in doubt.

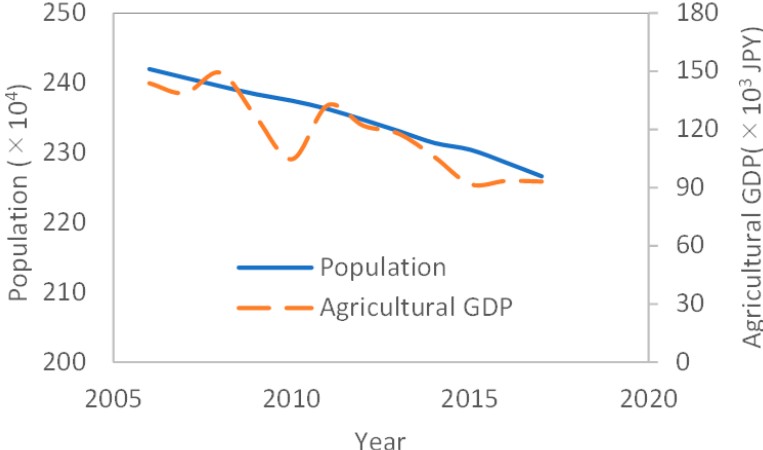

**Figure 5.** The population of Niigata and GDP of agricultural sector (Data source: [36,37]).

### 3.2. Case 2: Beijing-Tianjin-Hebei

Figure 6 shows the locations of Beijing, Tianjin, and Hebei Province. Beijing, Tianjin, and Hebei Province is the largest urban agglomeration of China, covering 2.2% of all land resources, accounting for 8% of the country's population, and generating 10% of the GDP in 2015 [38]. Ecological footprints of Beijing, Tianjin, and Hebei have been studied and found that the Hebei Province has both the highest value of ecological footprint and highest increase rate among the three regions [39,40]. However, these studies did not address the question that the Hebei province's ecological footprint was used for what. Aiming at synergistic development in the Beijing-Tianjin-Hebei region, the Beijing-Tianjin-Hebei Collaborative Development Strategy was proposed in 2014 as a national strategy for achieving a higher-quality economic growth, a more reasonable industrial structure, and a better ecological environment as well [41]. However, a recent publication [42] reported

that the consumption of natural resources in Hebei Province is still mainly reflected in the consumption of coal, petroleum-based energy, and the development model is still the same as it was in 2010.

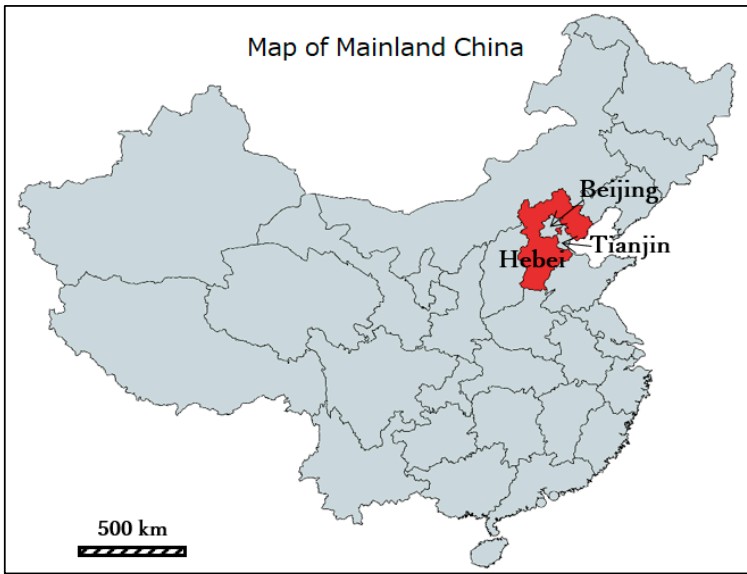

**Figure 6.** Locations of Beijing, Tianjin and Hebei.

Tianjin's urban area is the third largest in China but does not have sufficient water resources within its administrative boundary, due to natural geographic and climatic conditions. The average annual precipitation in Tianjin is about 550~680 mm. The water resources per capita is 160 m$^3$, which is 1/15 of the national average, and lower than the absolute water scarcity level defined by UN, which is 500 m$^3$ per capita. To solve the water scarcity problem, a project was implemented in 1981 to divert water from the Luan River to Tianjin and was completed in 1983 [43]. It supplies 1 billion m$^3$ of water to the city in a relatively dry year, which is about 65% of Tianjin's own total annual renewable water resources. This diversion stopped Tianjin's suffering from brackish water as a drinking water source and served as an engine for Tianjin's sustainable development. Nevertheless, the diversion project caused a significant change in the downstream reaches of the Luan River.

Luan River has its source in the province of Hebei and flows northwards into the province of Inner Mongolia, and then turns southeast back into Hebei to the river mouth on the Bohai Sea. The main channel of the waterway is 888 km long, with a catchment area of 44,900 km$^2$. Due to the diversion, the flow regime in the downstream of the Luan River has been significantly changed. The mean annual runoff reduced from $47.2 \times 10^8$ m$^3$ to $18.4 \times 10^8$ m$^3$ [44]. As the flow regime altered, so is the sediment transport. As a result, the sediment flux from the river to the sea decreased 483.6 times from 2000 to 2009 compared to 1950–1959, leading to severe coastal erosion [45]. Due to the reduction of flow and sediment supply, natural wetlands in the Luan River's delta area were degraded, and many of them were converted to farmlands, as shown in Figure 7. The conversion increased the water use for agriculture downstream. Consequently, the annual runoff into the sea from the river almost ceased since 2000 during the non-flood season in normal years. In 2003, the number of zero flow days exceeded 200. Reduced flow will lead to river-mouth clogging during normal days, affecting channel capacity to pass flood waters during flood season. In the case of Luan River, it was estimated that the channel conveyance has decreased from 1200 to 200~400 from 1956 to 1989. Thus, it is a case that water supply management to a city increased flood risk in the resource-providing region. Such an issue should be brought to forefront and given more analysis and discussion.

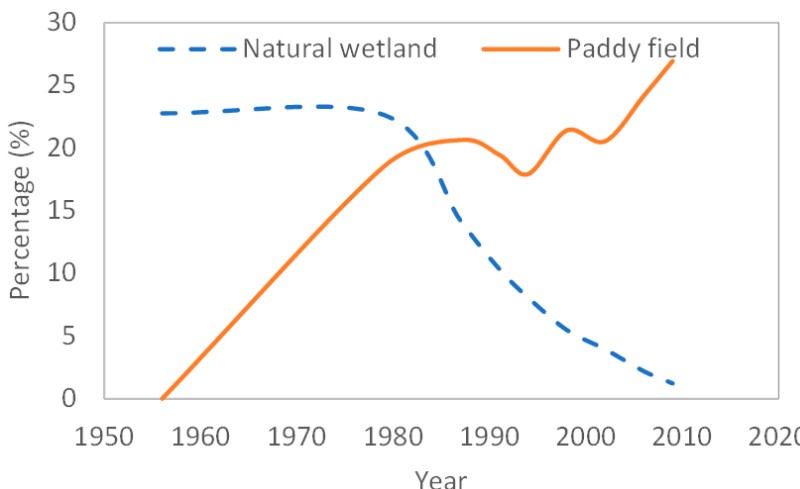

**Figure 7.** The percentages of natural wetlands and paddy fields in total wetland area in the delta of the Luan River (Data source: [46]).

Furthermore, saltwater intrusion increased from 7.98 km to 9.62 km around the river mouth because of the surface runoff reduction, causing high soil salinity [47]. Therefore, Tianjin's saltwater intrusion problem in the Bohai Coastal zone was shifted to the Luan River's estuary in Heibei. The sustainability of agriculture in the delta area of the Luan River is threatened accordingly.

Previous studies also indicated that the Hebei province is a big net virtual water exporter to Beijing through agricultural products [48]. The water resources per capita in Hebei Province is 307 m$^3$, 1/7 of the national average. The agricultural water uses in the Hebei province accounts for 75% of the total water use, while the agricultural water uses in Beijing decreased from 40.8% in 2000 to 31% in 2010 [49]. Due to high water demand and limited surface water resources, groundwater in Hebei Province has been overexploited, causing water table drop by more than 2 m annually [50].

Figure 8 shows the net population migration rates (difference between in-migration and out-migration) in Beijing, Tianjin, and Hebei Province. Like what is happening in Japan, rural areas in China lost their populations to large cities like Beijing and Tianjin, although China has a household registration system, which may have controlled population migration to a certain extent.

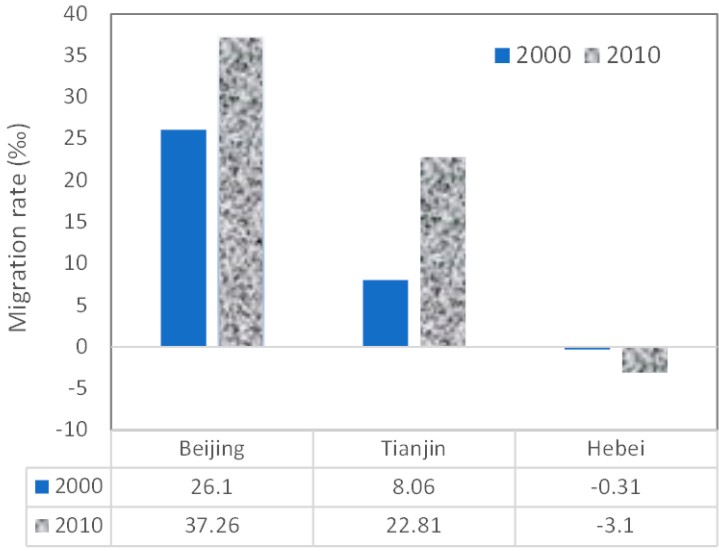

| | Beijing | Tianjin | Hebei |
|---|---|---|---|
| 2000 | 26.1 | 8.06 | -0.31 |
| 2010 | 37.26 | 22.81 | -3.1 |

**Figure 8.** Population migration rates in Beijing, Tianjin, and Hebei (Data source: [51,52]).

## 4. Discussion

Large cities' dependence on rural areas for both natural and human resources may cause negative externalities to rural areas and affect their sustainability. On the other hand, dependency can be considered as a vulnerability of large cities. Vulnerability represents the interface between threats to human well-being and the capacity of people and communities to cope with those threats [53]. When applied in unban context, it may be stated as representing the interface between threats to material and service supply chain and population safety and stability, and urban dwellers' capacity to cope with those threats. However, just knowing what can sustain a city's prosperity but without knowing what the cost is, and how the cost is shouldered, urban sustainability cannot be truly achieved on a global scale. Therefore, a new framework for urban sustainability is proposed, as depicted in Figure 9. The overlap area implies minimal externality, vulnerability, and instability.

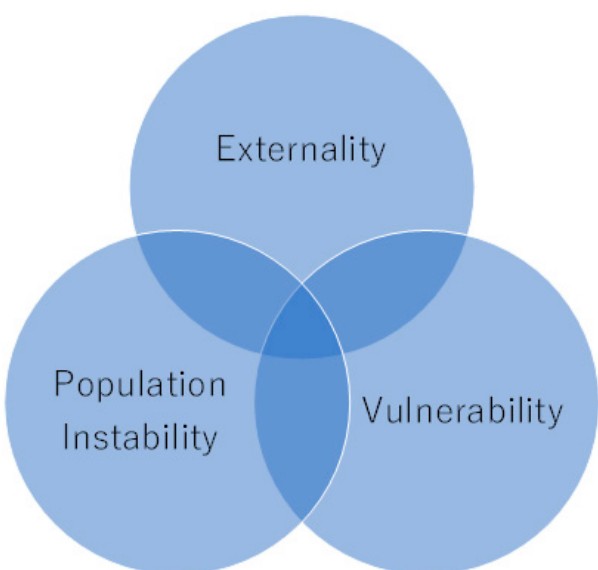

**Figure 9.** A new framework for sustainability study.

Like the conventional sustainability concept, it has three circles that intersect with each other. The three components are (1) Externality, (2) Vulnerability and (3) Population instability. The overlap area indicates minimal externality, vulnerability, and instability, implying that sustainability is being achieved.

This new framework is indeed advocating a paradigm shift. The rationale is that impacts of urban development on the local environment, such as pollution of local streams, have been intensively and extensively studied and well-recognized. However, externalities an urban development may have caused in regions away from the development site are still less analyzed and poorly understood. For example, the demand for wood in a coastal city may cause deforestation in a remote area and such connections are still insufficiently documented. Therefore, the emphasis of sustainability study in the environment dimension should now be placed on externality. Externalities could not be eliminated, and efforts to minimize them are key towards sustainability. On the other hand, if society may be easily affected by natural or human-caused disasters or disturbances, sustainability will be out of the question. Therefore, a vulnerability assessment is also crucial to sustainable development. Sustainability is impossible unless societal vulnerabilities, such as exposure or susceptibility to natural disasters, are minimized or reduced significantly. Furthermore, if a region's population is stable, economic activities will develop naturally to meet people's various needs. History tells that human beings can survive without a modern economy, but the modern economy could not be developed and sustained without maintaining a certain population level. It should be pointed out that vulnerabilities, not population size, hamper the economy of a populous and poor country. China is a good example.

Despite being the most populous country, its economic development is tremendous so far. However, for sustainable development, population migration between large cities and rural areas must be streamlined. Demographic dividend or the right population's age structure can lead to maximum productivity and minimum vulnerability if appropriately managed. Therefore, the three new pillars provide a platform for more focused and in-depth sustainability research.

To reduce externality, approaches such as payment for ecosystem services (PES) can be employed. In 2008, Japanese government started a hometown tax donation system, in which taxpayers can choose to divert part of their residential tax to a specified local government. It was mainly designed for people working in large cities to support their hometowns, reducing externalities. However, because small cities and towns that receive tax donations return gifts to donators, and the offerings vary significantly in value, the hometown tax donation became a gift-hunting game. Residents decide where to donate considering the monetary values of return gifts, not externalities their urban life may have caused.

## 5. Conclusions

Through case studies in Japan and China, evidence was provided to support the statement that a city's sustainability could be at the cost of unsustainability of rural areas. It highlights the issue of how to achieve global sustainability, leaving no one behind. For paradigm shift, a new framework was proposed for deepening sustainability studies with new focuses. Primarily, it advocates that population instability should be considered as an essential factor affecting sustainable development. Keeping the right size of the population at the right place, at the right time, and at the right cost will be a pathway towards sustainability. A further research question worth exploring is if enhancing green mobility could reduce one-way population flow.

**Funding:** This research received no external funding.

**Institutional Review Board Statement:** Not applicable.

**Informed Consent Statement:** Not applicable.

**Data Availability Statement:** Not applicable.

**Acknowledgments:** This work is part of Sophia Research Branding Project. The author would like to give thanks to three reviewers for their constructive comments.

**Conflicts of Interest:** The authors declare no conflict of interest.

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
