# Peer review of "Urban Sustainability at the Cost of Rural Unsustainability"

_sustainability, doi:10.3390/su13105466_

Round 1
Reviewer 1 Report
Please read my comments in a separated file

Author Response
Your professional comments are highly appreciated

Reviewer 2 Report
The premise makes sense and alludes to a whole line of research in geography on core-periphery relationships that the author should review
Should also review City Doughnut model framework that focuses on documenting local and global impacts on sustainable city performance
While the study shows correlation with population growth and decline in the paired cities, there is no clear evidence that it has led to declines in well-being in the rural area. Well-being is a sustainability measure (see the SDGs), population size is not.
The externalities argument has always has existed in urban studies and is a proper line of research especially with respect to natural resources. Urban metabolism studies emphasize this as well. Cities are almost always net importers of resources.
Good to expand and build upon ecological footprint work to require a focus on place-based impacts and to add a social footprint component.
Abstract: authors equate growth with spatial expansion; cities can grow economically or demographically without increasing spatial extent or sprawl; that is why sustainability movement looks to increase development in urban settings eg smart growth, New urbanism, infill development, increase density, etc
Abstract: depends on the notion of "prosperity" the idea of sustainability is that aspects of well being can increase without drawing negatively upon natural capital or negatively impacting external social communities. also shifts to service or knowledge economy has less impacts on rural communities
Materials and Methods: still missing methods. what and how were these data collected? How were case cities selected?
Page 4 lines 174-176 require deletion “ Since the conditions for 174 childrearing in Tokyo is not favorable, many women in Tokyo might not want to have children. Therefore, the concentration of female in the capital might cause the nationwide birth rate to decline further.” There is no evidence to suggest that the decline in population is due to out migration than to just lower birth rates; no evidence to suggest that out migrants destination is the Tokyo area. How would the presence of females lower birth rate! One would normally argue the opposite. If you wanted to make an argument along these lines you could explore sex ration changes.
Page 5 lines 272-274 the reduction in agricultural labor force is a global phenomenon due in part to increases agricultural intensity independent of migration. In fact, some have blamed increased ag intensity, production as a push factor leading to out migration
Page 6 line 274 utilizes GDP as response variable, but what about measures of agricultural yield and production? did that decline? That can give you measure of agricultural intensity which one can argue is a good thing
Page 7 lines 316-317: how can you call this sustainable development when you are at the same time highlighting the environmental damage of the diversion?
Page 7 line 336 on “increased flood risk” is jmissing explanation. how is this increasing flood risk?
Page 9 lines 438-443
- but population size is just a number not a condition. shouldn't the focus be on well-being? /does the population decline cause decline in well-being or other sustainability measures? reducing population volume/density in certain areas may free resources for the remaining population or reduce stress on environment. Shouldn't those be the ultimate indicators and not population size?
- “modern economy could not be developed and sustained without certain 441 level of population. yet that is not happening elsewhere which is why we have a push for "sustainable development" even in growing, stable or declining population settings.
- Lacking evidence to suppor the claims that “modern economy could not be developed and sustained without certain 441 level of population.”
- As author states, nations with large populations and poor are still acing vulnerabilities so it is not population size itself that raise concerns.
- Demographic Dividend: yes, so it is not necessarily population size or change that necessarily contributes to growing or prosperous economy
English language and grammar issues are widespread
Conclusion is sparse and weak
Author Response

(The authors gave the same response as above.)

Reviewer 3 Report
BRIEF SUMMARY – The work focuses on urban sustainability and the related fallouts on rural areas; based on the analysis of two case studies, it proposes a new framework for urban sustainability, built upon three pillars: Externalities, Vulnerabilities and Population stability.
BROAD COMMENTS – The topic, although not very new, is fully in line with the scope of the journal. The main research question is clearly stated and adequately introduced, although literature review should be considerably enriched. Given the importance of the topic addressed – urban sustainability – a specific ‘Literature review’ section could be arranged. Both sections 1 and 2 gather too briefly some very important points, that should deserve adequate space.
The choice of the method (‘case-study approach’) is explicitly motivated, but very few information is given as to the type of activities carried out. The selection of cases (the cities of Tokyo and Beijing) is adequately motivated.
Overall the works has major criticalities. In the first place, materials for the study are not insufficiently described, as to the data used, their time reference and validity as well as the limits of validity of the whole study. Also data sources are not always mentioned. Although the work proposed is a short communication and not a full research article, it should anyway make the author’s contribution clear in respect to the collected data. The border between source data and the subsequent elaboration is not clear: which are the starting data and what type of elaboration have they undergone? The reading just gives the idea of a plain data collection, used as case description. Then, it is not clear if data presented represent the context description or the result of an analysis. Also, without an explanation of data elaboration carried out, it is rather difficult to assure the replicability of the study.
While the work title refers to the relationship between cities and rural areas, in many points (52-53, 60-61, 95-96, 116) the text identifies, as major gap in current knowledge, the lack of information on the effects of urban growth on other surrounding cities. Then, the objective of the work remains rather uncertain and the reader gets confused about what the goal of the dissertation is.
Other criticalities relate to the basic assumptions of the work. In the first place, the choice to exclude economic issues in the case definition (in lines 91-99) is unclear; it should be better explained. More generally, the nature of the link between population dynamics and prosperity (or sustainability) appears somehow subjective and debatable. One example of that is represented by the Japanese capital: in the author’s words, the social and economic systems in Tokyo are built upon an increasing population hypothesis, so if population decreases, they risk to become socially unsustainable (138-141). This is anyway a specific policy decision for that city, and it does not necessarily imply that in the rest of the world an increase in population is a guarantee for prosperity or sustainability.
In Section 3.1 sustainability is identified and represented through the GDP indicator, and this can be to some extent a contradiction. The work seems to be entirely built on the equation “sustainability = increasing population”, or “sustainability = growing economy”, with little delving into the environmental implications of population growth or of economic development in terms of resource consumption; only the increase in water temperature (for Tokyo) and soil erosion (for Beijing) are mentioned.
Furthermore, the two cases are described rather heterogeneously (Tokyo: fertility rate, water, food and energy consumption; Beijing: water consumption, soil erosion, migration rate), which can be, on one hand, justified through differences in data availability, but confirming on the other hand that the work is for a large part a report of statistical data. These already show, by themselves, that Tokyo - for example - is largely dependent from the surrounding areas; what the author aims to get to, based on such consideration, is less clear.
The whole structure of the work, in terms of section organization, should be reconsidered, especially to clarify if the ‘result’ of the research is the description of cases or the framework proposed in the ‘Discussion’ section. Also the fact that the ‘Results’ are widely dependent on bibliographic references is rather uncommon.
In figure 6, data is very old (years 2000-2010), in respect to other data, without giving an explanation for that. This weakens the stress on population dynamics as a central point in the dissertation: if they are meant to be proposed as a key factor, their coherence (especially in time alignment) with the other data and indicators involved and across the two cases is more than crucial for reliable statements.
The ‘Discussion’ section is excessively short; a new framework based on externalities, vulnerability and population stability is proposed as a substitute for the conventional triple bottom line concept of environmental, social and economic sustainability. Nevertheless, a strong demonstration of the fact that those new components are as comprehensive is missing. In order to propose a new reference framework, the transition from case analyses to a generalization of concepts is necessary; on the contrary, many statements in the work appear arbitrary and closely connected to the case examined, e.g. in lines 442-443, 438-440, etc., while others remain unclear (lines 437-438: what is meant by ‘societal vulnerabilities’? Lines 427-433: is distance a reason sufficient to replace the concept of environmental impacts through the one of externalities? Resource consumption is already considered in environmental impact assessments, regardless of the distance to which it occurs. If the rationale of the work is substantially to suggest a shift from local to global, it should be stated more clearly).
Moreover, in order to build a new framework, a clear definition of the three components, the identification of sub-components and the proposal of possible indicators are also important. The three components assumed appear rather heterogeneous in nature; furthermore, in the conventional framework, sustainability is identified in the overlap area of environmental, social and economic sustainable developments: how is sustainability identified in the new framework? Is a city sustainable when externality, vulnerability and population stability occur simultaneously (Figure 7)?
Population stability is not in itself necessarily a requisite for sustainability; indeed, the proposed scheme does not consider resilience, that can enable a city or region to remain within sustainability limits supporting manifold changes, of which population dynamics are one possible example. The proposed framework can prove reliable in the case of Tokyo, for its peculiar policies based on population increase, but – in order to acknowledge a general validity to it as a reference - population stability as essential component of urban sustainability needs to be demonstrated in more explicit and rigorous ways.
Although the work is a communication and not an article, the ‘Conclusion’ section is really too brief and it reaffirms a widely shared concept (lines 451-452), still in probabilistic terms. The validity of the proposed framework is stated but not proven; the limits of the research reported and indications for future research are also missing.
Further clarifications on the positioning of the work in respect to the objectives of the ongoing research (Sophia Research Branding Project) could contribute to a better understanding of the proposed work.
My warm suggestion is to revise the whole work, clarifying what the real ‘Result’ of the work is, explaining the activities performed on data, and base the proposed framework on more general and rigorous motivations.
SPECIFIC COMMENTS
71: “indictor”; please check
112-113: unclear period; please, explain or rephrase
114: what does ‘it’ refer to?
118: “Cities without having ecological deficit is not guaranteed to be sustainable”; please check.
122: “resources use”; please check.
128-129: “the population movement as the pivotal to characterize the externality of urban development”; the relation between population movement and the externality of urban development should be explained, demonstrated and discussed.
174-175: “Since the conditions for childrearing in Tokyo is not favourable”; please check.
430-431: “such as demand for wood in a coastal city may cause deforestation in a remote area”; please consider rephrasing the period.
Figures 1 to 5: please add data sources.
Author Response

(The authors gave the same response as above.)

Round 2
Reviewer 3 Report
Although some points still remain debatable, the author's revised version addressed at least some of the many questions raised through complementary information and/or explanations. Literature, discussion, and conclusions were extended.
A limited improvement of the overall quality of the work can be acknowledged, sufficient to justify acceptance for publication.